# The Response Regulator VC1795 of *Vibrio* Pathogenicity Island-2 Contributes to Intestinal Colonization by *Vibrio cholerae*

**DOI:** 10.3390/ijms241713523

**Published:** 2023-08-31

**Authors:** Junxiang Yan, Qian Liu, Xinke Xue, Jinghao Li, Yuehua Li, Yingying Su, Boyang Cao

**Affiliations:** 1EDA Institute of Biological Sciences and Biotechnology, Nankai University, Tianjin 300457, China; 2Key Laboratory of Molecular Microbiology and Technology of the Ministry of Education, Nankai University, Tianjin 300457, China; 3Tianjin Key Laboratory of Microbial Functional Genomics, TEDA College, Nankai University, Tianjin 300457, China

**Keywords:** *Vibrio cholerae*, VC1795, CRP, intestinal colonization, pathogenicity

## Abstract

*Vibrio cholerae* is an intestinal pathogen that can cause severe diarrheal disease. The disease has afflicted millions of people since the 19th century and has aroused global concern. The *Vibrio* Pathogenicity Island-2 (VPI-2) is a 57.3 kb region, *VC1758*–*VC1809*, which is present in choleragenic *V. cholerae*. At present, little is known about the function of VC1795 in the VPI-2 of *V. cholerae*. In this study, the intestinal colonization ability of the Δ*VC1795* strain was significantly reduced compared to that of the wild-type strain, and the colonization ability was restored to the wild-type strain after *VC1795* gene replacement. This result indicated that the *VC1795* gene plays a key role in the intestinal colonization and pathogenicity of *V. cholerae*. Then, we explored the upstream and downstream regulation mechanisms of the *VC1795* gene. Cyclic adenylate receptor protein (CRP) was identified as being located upstream of *VC1795* by a DNA pull-down assay and electrophoretic mobility shift assays (EMSAs) and negatively regulating the expression of *VC1795*. In addition, the results of Chromatin immunoprecipitation followed by sequencing (ChIP-seq), EMSAs, and Quantitative Real-Time Polymerase Chain Reaction (qRT-PCR) indicated that VC1795 directly negatively regulates the expression of its downstream gene, *VC1794*. Furthermore, by using qRT-PCR, we hypothesized that VC1795 indirectly positively regulates the toxin-coregulated pilus (TCP) cluster to influence the colonization ability of *V. cholerae* in intestinal tracts. In short, our findings support the key regulatory role of VC1795 in bacterial pathogenesis as well as lay the groundwork for the further determination of the complex regulatory network of VC1795 in bacteria.

## 1. Introduction

*Vibrio cholerae* is a Gram-negative motile curved rod with a single polar flagellum, which causes severe diarrheal disease cholera [1]. Every year, there are 5–7 million cases of cholera worldwide [2]. *V. cholerae* predominantly resides in a variety of aqueous environments [3], and human infection normally starts with the ingestion of food or water containing the facultative pathogen. To cause disease in the small intestine, *V. cholerae* must assemble the colonization factor toxin-coregulated pilus (TCP) and secrete cholera toxin (CT), the major virulence factor [4,5]. The expression of the genes encoding TCP, CT, and other virulence factors is coordinately regulated by the ToxR regulon [6,7,8,9].

Four pathogenicity islands (PAIs) of *Vibrio*, the VPI-1, VPI-2, *Vibrio* seventh pandemic island-I (VSP-I), and VSP-II, were found in *V. cholerae* O1 and O139 [10,11,12]. It is well-known that the VPI-1 consists of the TCP gene cluster, the virulence regulators ToxT and TcpPH, and the accessory colonization factor (ACF) [13,14]. The VPI-2 encodes a P4-like integrase, a restriction-modification system (RM), a sialic acid metabolism region, a Mu phage-like region, and neuraminidase (VC1784), which is a glycosyl hydrolase known to release sialic acid [11]. Mu phage is known to cause spontaneous deletion and insertion events in chromosomal DNA. The variant VPI-2 regions among the O1 and O139 isolates are likely due to Mu phage deletion events. Jermyn et al. determined that all the examined toxigenic *V. cholerae* O1 serogroup isolates contained the VPI-2 of 57.3 kb, encoding 52 open reading frames (ORFs), whereas the non-toxigenic isolates lacked the island; also, the amino acid sequence of the ORFs *VC1794* and *VC1795* shared a similarity with the gp12 protein of the PSA bacteriophage and the Mor protein of the Mu phage, suggesting that these genes may be involved in the mobilization and integration of this region [11]. This is similar to our alignment results. VC1795 is a Mor transcription activator family protein. It is known that Mu is a temperate bacteriophage of *E. coli* K-12 and several other enteric bacteria [15]. There are three phases of transcription during the lytic development of bacteriophage Mu: early, middle, and late [16]. The initiation of the middle transcription from Pm requires the phage-encoded activator, Mor [15], which is a sequence-specific DNA-binding protein that promotes transcription activation via interactions with the C-terminal domains of the α and σ subunits of the bacterial RNA polymerase [17]. Kumaraswami M. et al. determined that the first structure for a member of the Mor/C family of transcription activators is composed of two domains, the N-terminal dimerization domain and the C-terminal DNA-binding domain, which are connected by a linker containing a β strand [17]. Our alignment results revealed that VC1795 is a Mor transcription activator family protein, contributes to mice intestinal colonization, and negatively regulates the expression of *VC1794*. More importantly, both *VC1794* and *VC1795* belong to a Mu phage-like region of VSP-II in the pathogenic *V. cholerae* O1 and O139 stains.

Cyclic adenylate receptor protein (CRP) is a typical bacterial global regulator, which was first identified in the process of catabolism in *E. coli* [18]. In *V. cholerae*, CRP (VC2614) was composed of 633 bases and 210 amino acids, with a protein size of 23.64 KDa and a GC content of 47.23%. After cAMP forms the cAMP–CRP polymer with its receptor protein CRP, it binds to TGTGA-N6-TCACA, the upstream conserved sequence of the promoter, and interacts with the RNA polymerase, thus regulating gene transcription [19,20,21]. In addition, the concentration of cAMP is very important for the transcriptional activity of CRP. The concentration of cAMP affects the conformation and biological properties of CRP, while in the promoter region of some genes, only the cAMP–CRP complex at a low cAMP concentration has transcriptional activity [22]. The cAMP–CRP complex is known to suppress the expression of *tcpP*/*tcpH* in *V. cholerae* [23]. In this study, we found that CRP directly negatively regulates the expression of *VC1795*.

The network of VC1795 in *V. cholerae* was described in this work, and the results revealed that *VC1795* is directly controlled by CRP; meanwhile, VC1795 directly negatively regulates the expression of *VC1794* and indirectly positively regulates the expression of the TCP cluster, thus affecting the virulence and colonization of *V. cholerae*.

## 2. Results

### 2.1. VPI-2 of Strain V. cholerae O1 EI Tor El2382

The VPI-2 gene cluster of *V. cholerae* El2382 is 56.8 kb in length and encompasses 50 open reading frames (ORFs). Each individual ORF of the VPI-2 gene cluster of *V. cholerae* N16961 was mapped, and it was found to stretch from *VC1758* to *VC1809*, including a P4-like integrase (*VC1758*), a restriction-modification system (*VC1765*–*VC1769*), a sialic acid metabolism region (*VC1773*–*VC1784*), and a Mu phage-like region (*VC1789*–*VC1809*). Also, there is a chromosomal insertion at a tRNA–serine (*VC1757*) locus that is flanked by direct repeats of *attL* and *attR* (Figure 1). Two ORFs of *VC1761* and *VC1787* were missed, so the VPI-2 gene cluster of *V. cholerae* El2382 is 500 bp shorter than that of N16961, which is 57.3 kb in length (Appendix A).

### 2.2. *Δ*VC1795 Isogenic Mutant Attenuates V. cholerae Virulence against Infant Mouse Intestinal Colonization

In order to understand the physiological function of the *VC1795* gene in *V. cholerae* O1 EI Tor E12382, the Δ*VC1795* isogenic mutant was generated. We measured and found that the colonization ability of Δ*VC1795* was reduced 2.74-fold compared to the WT (Figure 2). Moreover, we constructed the complemented strain (Δ*VC1795* + p*VC1795*) containing a functional copy of the *VC1795* sequence using the plasmid pBAD33 and found that the colonization ability was restored to the WT level upon complementation (Figure 2). These results confirmed that the Δ*VC1795* isogenic mutant attenuates the colonization ability of the infant mouse intestine.

### 2.3. VC1795 Is Directly Represses by cAMP–CRP Complex

After the pull-down experiment was completed, SDS-PAGE electrophoresis was performed on the eluted proteins (Appendix A). The bands were cut off and sent to mass spectrometry for analysis. The results are shown in Appendix A. It was speculated that three regulators, VC1087, CRP, and CytR, may be the upstream regulatory proteins of VC1795. To verify whether *VC1795* is regulated by VC1087, CRP, and CytR as predicted, the isogeneic deletion mutants of Δ*VC1087*, Δ*crp*, and Δ*cytR* were constructed. qRT-PCR was performed, which revealed that the transcription of *VC1795* increased by approximately 2.8-fold in the Δ*crp* mutant compared with that in the WT strain under AKI induction conditions, while it was unchanged in the other two (Figure 3A). Next, CRP protein with a His-tag was purified, and an EMSA was carried out. The results indicated that CRP could directly bind to the *VC1795* promoter (Figure 3B). Furthermore, EMSAs were carried out between CRP protein and 4.5s RNA as a negative control (Figure 3C). These data revealed that CRP is a negative regulator of *VC1795*.

### 2.4. VC1974 Is the Downstream Target of VC1795

To investigate the effect of VC1795 downstream genes on *V. cholerae* pathogenicity, ChIP-seq was performed. A motif was generated based on all the ChIP peaks using the MEME program, and the height of each letter represented the occurrence frequency at each location (Figure 4A). Subsequently, according to the results of ChIP-seq, we selected seven genes for qRT-PCR validation, which revealed that the transcription of *VC1794* increased by approximately 3.4-fold in the Δ*VC1795* mutant compared with that in the WT strain under AKI induction conditions (Figure 4B). And VC1795 protein with a His-tag was purified, and EMSAs were carried out; the results indicated that VC1795 could directly bind to the *VC1794* promoter (Figure 4C). Moreover, EMSAs were carried out between VC1795 and 4.5s RNA as a negative control (Figure 4D). These data revealed that VC1795 is a negative regulator of *VC1794*.

### 2.5. *Δ*VC1794 Isogenic Deletion Mutant Increased Virulence with Infant Mouse Intestinal Colonization

To address the role of VC1794 in *V. cholerae* El2382 virulence, we compared the WT with its isogenic Δ*VC1794* deletion mutants for the ability to colonize the infant mouse intestine 24 h post-inoculation. While the WT strain colonized the intestine at a level of 3.8 × 10^4^ CFU/mL, the Δ*VC1794* mutant colonized the intestine at a level of 2.5 × 10^5^ CFU/mL, a 6.6-fold increase in the median number of CFU/mL in the mutant. The Δ*VC1794* mutant showed significantly increased colonization in the infant mouse intestine (Figure 5).

### 2.6. VC1795 Positively Regulates the TCP Cluster Associated with Colonization of V. cholerae

It was reported that TCP is the intestinal colonization factor of *V. cholerae*. By qRT-PCR, compared with WT strains, the *tcp* cluster expression of the strain with Δ*VC1795* significantly decreased, which shows that VC1795, via the *tcp* cluster, influences the colonization ability of *V. cholerae* in intestinal tracts (Figure 6).

### 2.7. Schematic of the Proposed VC1795 Regulatory Mechanism

In the present study, we revealed the VC1795-controlled pathways, and our proposal regarding the pathways regulated by VC1795 in *V. cholerae* is outlined in Figure 7.

## 3. Discussion

The two biotypes of *V. cholerae* O1 strains are known as classical and El Tor. While the first six cholera pandemics were caused by the classical biotype, the current seventh pandemic is due to the El Tor biotype [24]. The primary virulence genes in *V. cholerae* are pathogenicity islands (PAIs), including the *Vibrio* pathogenicity island 1 (VPI-1), VPI-2, VSP-1, and VSP-2, and *tcpA*, *toxR*, *T6SS*, *ctxA*, *hlyA*, *ctxB*, *zot*, *alsD*, *ace*, *rtxA*, *makA*, and *ompU* [25]. The evolution of pathogenic bacteria depends on PAIs, which are mobile integrated genetic elements (MIGEs) that include a wide variety of virulence factors. Most typically at a tRNA locus, PAIs integrate into the host genome of bacteria through integrase-mediated site-specific recombination [26]. The VPI-1 and VPI-2 are crucial for *V. cholerae* host colonization. The VPI-2 is 57.3 kb in length in the *V. cholerae* N16961 genome, consisting of 52 open reading frames (ORFs), *VC1758*–*VC1809* (Appendix A) [11]. Interestingly, we found that the VPI-2 gene cluster of *V. cholerae* El2382 is 500 bps shorter than the one of N16961 due to the loss of *VC1761* and *VC1787* (Figure 1). The functions of *VC1761* and *VC1787* in *V. cholerae* are yet unknown; however, given our incomplete understanding of the pathogenesis of *V. cholerae*, genes within the VPI-2 may play a role in virulence.

VC1795 is a Mor transcription activator family protein, by amino acid sequence alignment, of a Mu phage-like region (*VC1789* to *VC1809*) in *V. cholerae* El2382 (Appendix A). In this study, we found that the absence of *VC1795* caused the intestinal colonization ability to be significantly reduced compared with WT, which indicated that VC1795 plays a key role in the virulence of *V. cholerae*. Furthermore, we also revealed that CRP is an upstream regulator of *VC1795* and negatively regulates *VC1795* expression. The CRP binding sites were predicted in the promoter regions of the *VC1795* gene by using Virtual Footprint 3.0 (see Appendix A). Meanwhile, Kovacikova G et al. demonstrated that the detrimental impact of cAMP–CRP on the expression of virulence genes is due to its capacity to alter the AphA- and AphB-dependent transcriptional activation of t*cpPH* under a variety of circumstances [27]. In addition, many studies have also found that cAMP–CRP could affect the virulence and colonization of *V. cholerae* through different regulatory ways. For example, according to Zou M. et al., DegS regulates *V. cholerae*’s chemotaxis and motility through the cAMP–CRP–RpoS–FlhF pathway, which has an impact on the colonization of suckling mice intestines [28]. Muzhingi I et al. demonstrated that *V. cholerae* controls the pathogenicity of interactions with arthropod hosts via the activity of the two-component CrbS/R system and cAMP–CRP activates the expression of the *crbS* and *crbR* genes [29]. The direct negative regulation of VC1795 by cAMP–CRP found in this study is likely to reveal another pathway by which CRP affects the virulence of *V. cholerae*. Certainly, CRP may not be the only regulator upstream of *VC1795*, so the search for other regulators is our focus.

In addition, we found the downstream regulatory gene *VC1794* of VC1795. And VC1795 directly negatively regulates the expression of *VC1794*. The Δ*VC1794* isogenic deletion mutant increased virulence with infant mouse intestinal colonization, contrary to the results of Δ*VC1795*, and VC1794 was negatively correlated with the virulence of *V. cholerae*. A previous study indicated that the amino acid sequence alignment of VC1794 was similar to the gp12 protein of the PSA bacteriophage [11]. There were a few restrictions on this study though. Since both *VC1795* and *VC1794* are located in a Mu phage-like region of the VPI-2, the genes in this region are only present in the pathogenic *V. cholerae* strains. Prior to our study, no research was reported on *V. cholerae* that shows the function and virulence of VC1795 and VC1794. Meanwhile, the comprehensive regulatory role of VC1795 in *V. cholerae* is currently unknown as a regulatory factor, but we anticipate confirming this in the future through the RNA sequencing of the WT and the Δ*VC1795* strain.

In the study of *V. cholerae*, the numbers of specific functional genes were reported, but there are also many unknown genes, some of which are defined as “hypothesized regulatory proteins”. These may play a role in the pathogenesis of *V. cholerae* as a bridge between global regulatory proteins and virulence factors. The study of hypothesized regulatory proteins can not only enrich the regulatory mechanism of *V. cholerae* but also provide a theoretical basis for the development of drug targets and the study of precautions and therapies.

## 4. Material and Methods

### 4.1. Bacterial Strains, Plasmids, and Growth Conditions

*V. cholerae* O1 El Tor biotype strain El2382 was provided by Shanghai Municipal Center for Disease Control and Prevention and was isolated in 1994. In this study, all strains and plasmids used are listed in Appendix A. Bacterial strains were grown in Luria-Bertani (LB) broth with aeration at 37 °C or on LB agar plates at 37 °C. Antibiotics (Sigma-Aldrich, St. Louis, MO, USA) were used at the following concentrations: 30 μg·mL^−1^ polymyxin B was used for culture of *V. cholerae*; 25 μg·mL^−1^ chloramphenicol for screening and culturing of complemented strains containing the pBAD33 recombinant vector; 50 μg·mL^−1^ kanamycin for screening and culturing of protein expression strains containing the pET28a recombinant vector.

### 4.2. Construction of Mutant Strains and Complementation

In this study, in order to investigate the function and inter-relationship of genes, the corresponding gene deletion mutants were constructed with the suicide plasmid pRE112 (sucrose-sensitive lethal) method, which uses the principle of homologous recombination [30]. Briefly, genomic DNA was used as the template, and the upstream and downstream sequences of the target genes (*VC1795*, *crp*, *cytR*, *VC1807*, and *VC1794* were the five genes deleted in this study) were amplified by PCR using two pairs of primers: F1 and R1; F2 and R2 (Appendix A), respectively. Furthermore, F1 and R2 contain the restriction sites (SacI and KpnI were used by F1 and R2 of *VC1795*, *cytR*, *VC1807*, and *VC1794*, whereas KpnI and SmaI were used by F1 and R2 of *crp*) carried by the pRE112 plasmid, and F2 and R1 contain reverse complementary sequences within 10 bp of each other in their respective 5′ ends. The upstream and downstream sequences were linked together by overlap-extension PCR using F1 and R2, digested by the above restriction enzymes, and ligated into the pRE112 plasmid before being transferred to *E. coli* S17-1 λpir, which was conjugated with the WT strain. The mutant strains that successfully became a homologous recombinant were selected on LB agar plates containing 10% sucrose and 25 μg·mL^−1^ chloramphenicol (carried by the pRE112 plasmid). A schematic illustration of the deletion of all genes in this study is shown in Appendix A. And, for complementation, the reconstructed vector pBAD33 containing the target gene (VC1795) was electroporated into the isogenic Δ*VC1795* deletion mutant [31]. PCR and DNA sequencing were used to confirm the deletion mutants and complementation strains. In addition, all the primers for the identification and selection shown in Appendix A were completed using Primer Premier 5 software and fully complied with the design principles of the primers.

### 4.3. RNA Isolation and Quantitative Real-Time Polymerase Chain Reaction (qRT-PCR)

To explore the regulatory relationship between regulators and downstream genes, qRT-PCR was performed on the WT and deletion mutant strains. Briefly, the bacterial strains were grown in sterile media at 37 °C overnight with shaking. The overnight bacterial solution was transferred the following day at a 1:100 ratio to fresh medium until the bacteria had reached an OD600 of 0.6. Cells were collected by centrifugation at 4 °C, 8000× *g* for 5 min. Total RNA was isolated using TRIzol Reagent (Invitrogen, Waltham, MA, USA), according to the manufacturer’s protocol, followed by treatment with an RNase-Free DNase and dissolution in RNase-Free water. The RNA’s purity was assessed using a NanoDrop-2000 spectrophotometer (Thermo Fisher, Waltham, MA, USA). cDNA synthesis was performed using a PrimeScript RT reagent kit (Takara Bio, Shiga, Japan) with 1.2 μg total RNA in each reaction mix. The qRT-PCR reaction system was carried out in a total volume of 20 μL containing 200 nM of each primer, 1 μL of cDNA, and 10 μL of universal SYBR Green Master mix. qRT-PCR analysis was conducted on an Applied Biosystems ABI 7500 sequence detection system (Applied Biosystems, Foster City, CA, USA); meanwhile, the 16S rRNA gene was employed as a reference control, and the cycle threshold approach (2^−ΔΔCT^) [32] was used to compute the relative target gene expression levels as fold changes. Each experiment was carried out in triplicate.

### 4.4. Electrophoretic Mobility Shift Assays (EMSAs)

To further investigate the interactions between the regulators and the proteins in this work, we further implemented Electrophoretic Mobility Shift Assays to verify them in vitro. EMSAs are a technique that can be used to study the interaction between DNA-binding proteins and DNA-fragment-binding sequences. Based on the principle that DNA–protein complexes have different mobilities in polyacrylamide gel electrophoresis (PAGE), when the transcription factor binds to specific DNA, its mobility in PAGE is less than the DNA of an unbound protein, demonstrating the direct regulatory relationship between regulatory factors and DNA. The proteins for EMSAs in this study, CRP and VC1795, were cloned into pET-28a, expressed in *E. coli* BL21 (DE3), and purified using a Ni-NTA-Sefinose Column (Sangon Biotech, Shanghai, China). The promoter of the downstream gene of CRP, *VC1795*, and the promoter of the downstream gene of VC1795, *VC1794*, were amplified by PCR. Binding reactions were performed using previously described methods [33], DNA fragment (50 ng) of *VC1795* with purified CRP in 20 μL of binding buffer (50 mmol/L Tris-HCl (pH 8.3), 0.5 μmol/L cAMP, 1 μg Poly (dI.dC), 2.5 mmol/L EDTA, 250 mmol/L KCl, 5 mmol/L MgCl_2_, and 2.5 mmol/L DTT) at room temperature for 30 min. The DNA fragments of *VC1794* were incubated with VC1795 proteins in a 20 μL binding buffer (20 mM HEPES (pH 7.6), 150 mM KCl, 2.5 mM DTT, 10 mM (NH_4_)_2_SO_4_, 2.5 mM EDTA, and 5% glycerol) at room temperature for 30 min. DNA–protein complexes were separated by 6% PAGE in 0.5×TBE buffer and stained with Gel Red (Biotium, Fremont, CA, USA). Protein signal was visualized using a Typhoon phosphorimager (GE Healthcare, Pittsburgh, PA, USA).

### 4.5. Chromatin Immunoprecipitation Followed by Sequencing (ChIP-seq)

Chromatin immunoprecipitation (ChIP), also known as binding site analysis, is a method to study the interaction between proteins and DNA, while CHIP-seq is a combination of chromatin immunoprecipitation and high-throughput sequencing. In simple terms, the principle is to express a large number of target proteins with the 3 × Flag tag in bacteria while using the target proteins and the special sequences of the promoter regions of the target genes to bind to each other. Subsequently, an anti-3 × FLAG antibody that specifically binds to target proteins carrying 3 × Flag and protein A magnetic bead that specifically adsorbs the anti-3 × FLAG antibody that binds the target protein with 3 × Flag was added to enrich the downstream genes of the target proteins. In this study, in order to identify the downstream genes regulated by VC1795 in *V. cholerae*, ChIP-seq was performed as previously described [34], with some modifications to the protocol. Briefly, the pBAD33-VC1795-3 × Flag recombinant plasmid was constructed and electroporated into the Δ*VC1795* strain and then cultured until OD600 reached 0.6–0.8, and formaldehyde solution with a working concentration of 1% was added for 25 min at room temperature. By adding 0.5 M glycine, crosslinking was stopped. The samples were washed three times with ice-cold PBS before being centrifuged at 13,000 rpm for 15 min at 4 °C. The samples were then resuspended and sonicated to generate 200~500 bp DNA fragments. The samples were then centrifuged at 14,000 rpm for 15 min at 4 °C, and the supernatant was collected by using an anti-3 × FLAG antibody (Sigma-Aldrich, St. Louis, MO, USA) and protein A magnetic beads (Invitrogen, Waltham, MA, USA). The negative control (mock) was an aliquot with no additional antibodies. Samples were treated with 10 μL of 10 mg/mL proteinase K for 1 h at 55 °C and 10 μL of 10 mg/mL RNaseA for 1 h at 37 °C. A PCR purification kit (Qiagen, Germantown, MD, USA) was used to purify the DNA fragments. Novogene, Inc. constructed and sequenced the next-generation sequencing library after extracting the VC1795-ChIP and mock-ChIP DNA (Appendix A). The enriched peaks were identified using MACS software [35], and the VC1795-binding motif was generated using MEME analysis [36].

### 4.6. DNA Pull-Down Assay

In this study, to investigate the upstream regulators of *VC1795* and enrich the regulatory network of VC1795 in *V. cholerae*, DNA pull-down assay was performed as previously described [37], with some modifications. DNA pull-down assay is a technique for studying DNA and protein interactions in vitro. In short, biotin-labeled DNA fragments bind to streptavidin magnetic beads and incubate with nuclear proteins to purify proteins that interact with DNA fragments. The protein products obtained by washing and elution can be identified by mass spectrometry to screen out protein information that may interact with DNA fragments. In this work, DNA probes with biotin at the *VC1795* promoter 5′ end were generated by PCR using primer pairs (Appendix A). A NanoDrop 2000 spectrophotometer was used to analyze the concentration and quality of the probes. Total proteins were analyzed using 12% SDS-polyacrylamide gel electrophoresis (SDS-PAGE). Streptavidin Dynabeads were rinsed before being suspended in buffer A (50 mM Tris-HCl (pH 7.5), 1 M NaCl, 1 mM DTT, and 0.5 mM EDTA). The DNA probes were incubated with the beads at room temperature for 30 min. Following that, the DNA–bead complex was washed three times with buffer A and buffer B (20 mM Tris-HCl (pH 8.0), 100 mM NaCl, 1 mM DTT, 1 mM EDTA, and 10% glycerol). The magnetic particles were rinsed three times with buffer B after incubating with the cell extracts for 1 h at room temperature with shaking at 200 rpm. Following the elution of the DNA-binding proteins using elution buffer C, which contained varying amounts of NaCl (50 mM Tris-HCl (pH 8.0) and 200 mM, 500 mM, or 1 M NaCl), DNase I was then added to the reaction mixture. SDS-PAGE was used on the fractions that had been eluted. After that, a 5% acetic acid solution was added to stop the process. The protein bands of interest were excised from the gel with a scalpel and sent to mass spectrometry.

### 4.7. Intestinal Colonization Assay

In order to characterize the effects of *VC1795* and *VC1794* on the virulence of *V. cholerae*, an in vivo intestinal colonization assay was performed as previously described, with some modifications [38]. Laboratory animal CD-1 infant (5-day-old) mice used in this study were purchased from Beijing Vital River Laboratory Animal Technology Co., Ltd. (Beijing, China). Briefly, the WT, Δ*VC1795*, and Δ*VC1795* + p*VC1795* strains were cultured in LB medium overnight. The next day, the overnight bacterial solution was transferred to fresh LB at a 1:100 ratio until the bacteria were grown to an OD_600_ of 0.6, and the colony forming units (CFU) were counted. A total of 1 × 10^7^ CFUs for the WT, Δ*VC1795*, and Δ*VC1795* + p*VC1795* strains were collected and used in the gavage experiment of the CD-1 infant mice. The animal model was divided into three groups; 10 CD-1 infant mice in each group were fed. Following that, all the surviving mice were dissected, and the small intestine was removed and resuspended in 1 mL of sterile PBS to release all bacterial cells. Serial dilutions of the homogenized organs were plated on LB agar with the addition of 30 μg·mL^−1^ of polymyxin B and incubated at 37 °C for 24 h. In addition, the colonization ability of the Δ*VC1794* strain was also tested in accordance with the above experimental methods.

### 4.8. Statistical Analyses

Data were analyzed using GraphPad Prism v7.0 software (GraphPad Inc., La Jolla, CA, USA) [39]. All data are expressed as means ± standard deviation (SD). Differences between two groups were evaluated using two-tailed Student’s *t*-test or Mann–Whitney U test. Values of *p* ≤ 0.05, 0.01, or 0.001 were considered to be statistically significant (*), highly significant (**), or extremely significant (***), respectively. NS indicates not significant.

## Figures and Tables

**Figure 1 ijms-24-13523-f001:**
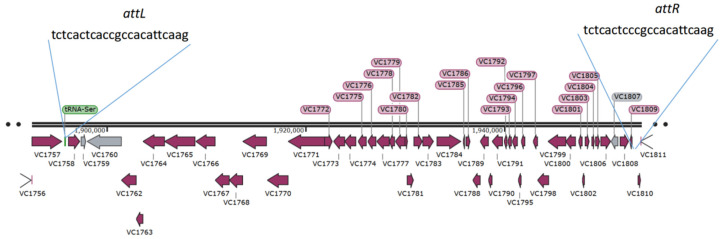
Schematic representation of VPI-2 (*VC1758* to *VC1809*) from *V. cholerae* strain El2382. The *attL* and *attR* genes indicate left and right attachment sites.

**Figure 2 ijms-24-13523-f002:**
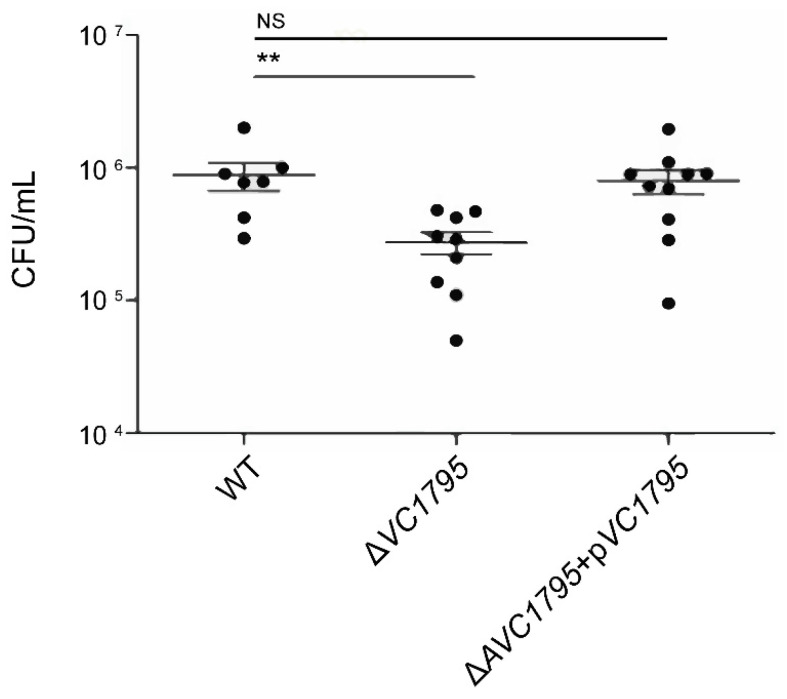
The Δ*VC1795* mutant attenuates the infant mice intestinal colonization ability of *V. cholerae*. The ability of different derivatives of WT, Δ*VC1795* mutant, and its complemented strain in *V. cholerae* El2382 to colonize infant mice intestines was analyzed. Significant differences are indicated by asterisks (** *p* ≤ 0.01); NS indicates not significant.

**Figure 3 ijms-24-13523-f003:**
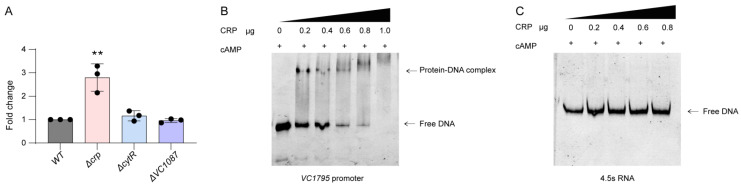
CRP directly negatively regulates the expression of *VC1795*. (**A**) The mRNA level of *VC1795* among the WT, Δ*crp*, Δ*cytR*, and Δ*VC1087* mutants. (**B**) EMSAs were carried out between CRP protein and *VC1795* promoter, with the concentration of CRP-His6 gradually increased. The amount of promoter DNA used in each reaction was 50 ng, and the binding buffer of EMSAs contained cAMP. (**C**) EMSAs were carried out between CRP protein and 4.5s RNA as a negative control. Significant differences are indicated by asterisks (** *p* ≤ 0.01).

**Figure 4 ijms-24-13523-f004:**
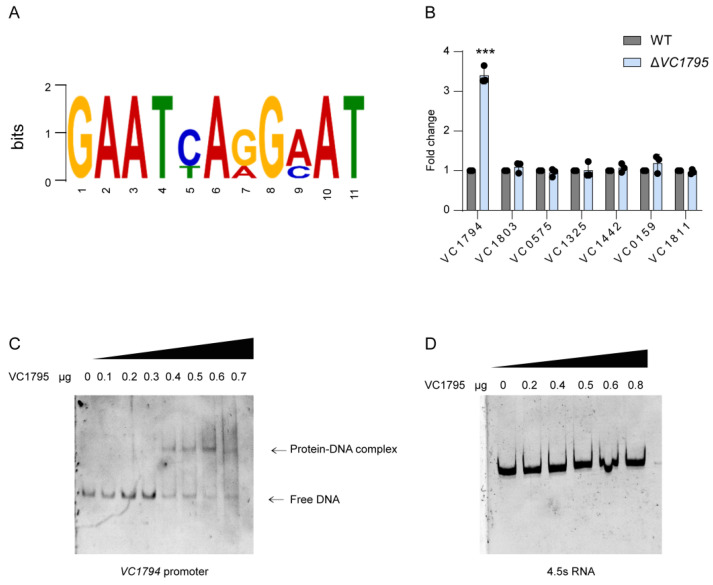
VC1795 directly negatively regulates the expression of *VC1794*. (**A**) The most significant motif identified by ChIP-seq using the MEME tool is shown. All peaks were used to define the binding motif. The height of each letter presents the relative frequency of each base at different positions in the consensus sequence. (**B**) The mRNA level of *VC1794*, *VC1803*, *VC0575*, *VC1325*, *VC1442*, *VC0159*, and *VC1811* of the WT and Δ*VC1795* mutant. (**C**) EMSAs were carried out between VC1795 protein and *VC1794* promoter, with the concentration of VC1795-His6 gradually increased. The amount of promoter DNA used in each reaction was 50 ng. (**D**) EMSAs were carried out between VC1795 protein and 4.5s RNA as a negative control. Significant differences are indicated by asterisks (*** *p* ≤ 0.001).

**Figure 5 ijms-24-13523-f005:**
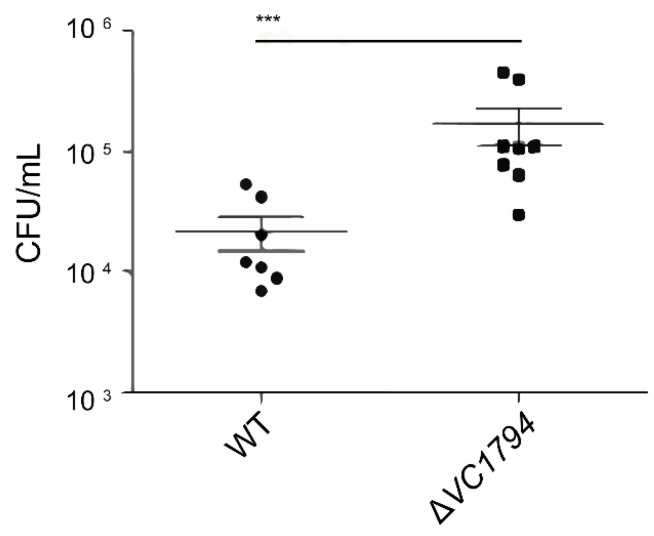
The Δ*VC1794* mutant attenuates the infant mice intestinal colonization ability of *V. cholerae*. The ability of different derivatives of WT and Δ*VC1794* mutant to colonize infant mice intestines were analyzed. The median is indicated by horizontal bars, and significant differences are indicated by asterisks (*** *p* ≤ 0.001).

**Figure 6 ijms-24-13523-f006:**
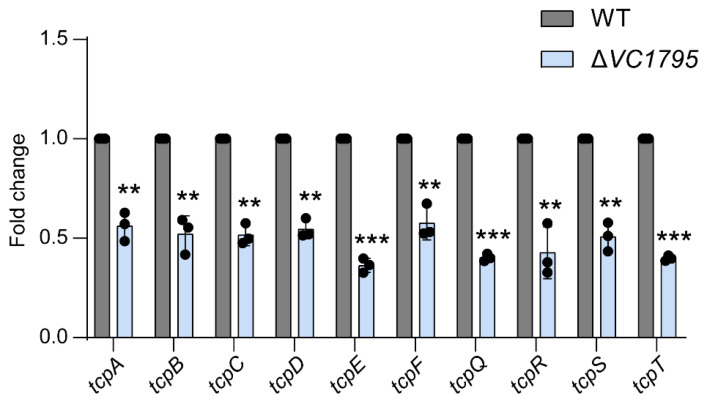
VC1795 functions as a positive regulator control the expression of *tcp* cluster. The mRNA level of *tcpA*, *tcpB*, *tcpC*, *tcpD*, *tcpE*, *tcpF*, *tcpQ*, *tcpS*, *tcpT*, and *tcpR* of the WT and Δ*VC1795* mutant. Significant differences are indicated by asterisks (*** *p* ≤ 0.001; ** *p* ≤ 0.01).

**Figure 7 ijms-24-13523-f007:**
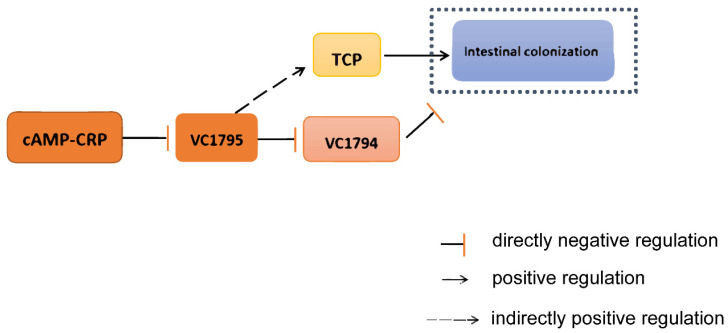
Schematic of the proposed VC1795 regulatory mechanism in *V. cholerae* El2382. The potential regulatory pathways and interplays of VC1795 are proposed according to our observations.

## Data Availability

All data are presented within the manuscript and the Appendix A.

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
