# Peer review of "The Response Regulator VC1795 of *Vibrio* Pathogenicity Island-2 Contributes to Intestinal Colonization by *Vibrio cholerae"

_ijms, 2023, doi:10.3390/ijms241713523_

Round 1

Reviewer 1 Report

See attached

Author Response

Reply Letter

Manuscript ID: ijms-2552480

Manuscript Title: The response regulator VC1795 of the VPI-2 contributes to intestinal colonization by Vibrio cholerae

Dear reviewer

Thank you very much for your valuable effort in handling this manuscript, ijms-2552480. We are also very grateful to you for your constructive comments, which helped us improve the quality of this research paper.

We hereby resubmit a revised manuscript to IJMS after making revisions and carefully considering all the comments. On the following pages, the detailed point-by-point responses to each comment are provided.

Thank you very much for your continued consideration of our work.

Best Regards,

Boyang Cao, Professor

Nankai University

Overview from reviewer 1:

Cholera, a severe diarrheal disease, is caused by the bacterium Vibrio cholerae. Successful intestinal colonization by this bacterium is crucial for causing infection and disease. This research investigates the response regulator VC1795 and its role in colonization, shedding light on how V. cholerae interacts with the host intestine. The study reveals that the VC1795 gene plays a pivotal role in intestinal colonization and the pathogenicity of V. cholerae. Through DNA pull-down and electrophoretic mobility shift assays (EMSAs), CRP was identified as an upstream regulator of VC1795, negatively impacting its expression. While the article is well-written and the results are interesting, there is room for improvement (major revisions) before publication. The following comments offer guidance for enhancing the manuscript.

Major Comments:

1 Introduction 

The introduction requires improvement, particularly in terms of framing the research question. While the authors have outlined the functional characteristics of the Cyclic Adenylate Receptor Protein (CRP) and the Pathogenicity Islands (PAIs) of Vibrio, they have not adequately justified the study's relevance. Although they focus on the functions of VC1795 in VPI-2 of V. cholerae, they have omitted referencing recent data related to the issue, which is crucial for establishing the study's rationale. A comprehensive overview of recent research highlighting the significance of VC1795 within the context of V. cholerae's intestinal colonization and its implications for cholera pathogenesis is lacking.

Our response: Thank you for the suggestions. We modified the narrative of the correlation between CRP and PAIs in the introduction section as ‘‘ Four pathogenicity islands (PAIs) of Vibrio, VPI-1, VPI-2, Vibrio seventh pandemic island-I (VSP-I), and VSP-II were found in V. cholerae O1 and O139 [10-12]. It's well known that VPI-1 consists of the toxin-coregulated pilus (TCP) gene cluster, the virulence regulators ToxT and TcpPH, and the accessory colonization factor (ACF) [13, 14]. The cAMP-CRP complex is known to suppress expression of tcpP/tcpH in V. cholerae [23]’’. Additionally, we found that CRP directly negatively regulates the expression of VC1795 which belongs to a Mu phage-like region of VPI-2. All the above data suggest that there is some correlation between CRP and PAIs and that it is possible to influence V. cholerae virulence through PAIs. The relevant explanation was added in page 3, lines 52-56; page 5, lines 92-93; and page 21, lines 387-389 of the revised manuscript.

Meanwhile, thank you for another suggestion. Prior to our study, nearly all information regarding the function and virulence of VC1795 in V. cholerae was not reported. However, we provide a brief overview of recent studies on the Mor transcription activator. The relevant explanation was added in page 4, lines 66-81 of the revised manuscript.

2 Discussion ·

Integrate findings from recent experimental studies that have delved into the role of VC1795 in V. cholerae's proficiency to colonize the intestinal mucosa. Furthermore, it is advisable to include a dedicated section outlining the limitations of the study. This addition will enhance the manuscript by presenting a well-rounded perspective on the research's scope and potential constraints.

Our response: Thanks for your suggestion. The relevant explanation was added in page 21, lines 398-404 of the revised manuscript.

3 Material and Methods ·

What did those antibiotics “polymyxin B 30 μg·mL-1; chloramphenicol, 25 μg·mL-1; kanamycin, 50 μg·mLl-1” have been used for?  

Our response: Thanks. 30 μg·mL-1 polymyxin B was used for culture of V. cholerae; 25 μg·mL-1 chloramphenicol for screening and culturing of complemented strains containing the pBAD33 recombinant vector; and 50 μg·mL-1 kanamycin for screening and culturing of protein expression strains containing the pET28a recombinant vector. The relevant explanation was added in page 6, lines 106-110 of the revised manuscript.

How were the primers utilized in the study identified and selected? ·

Our response: Thank you. In this study, the identification and selection of all the primers, shown in Table S2, were completed using the Primer Premier 5 software and fully complied with the design principles of the primers. The relevant explanation was added in page 7, lines 126-129 of the revised manuscript.

2.1. Intestinal colonization assay:

Our response: Thanks for your suggestion. The relevant explanation was added in page 10, lines 203-214 of the revised manuscript.

Minor Comments : 

In general, font sizes are not consistent throughout the manuscript, leading to difficulties in readability. Please standardize the font sizes and thoroughly proofread for consistency.

1 Introduction

Paragraph 2 ‘‘The cyclic AMP (cAMP)-cAMP receptor protein (CRP) ……..suppress expression of tcpP/tcpH in V. cholerae [15]’’: Adjust font and size

Our response: Adjusted.

2 Material and Methods 

Section 2: Material and Methods: Delete “s” from Materials

Our response: Changed. 

2.7, first sentence: in vivo should be in italics

Our response: Changed. 

Table 1: Genotype or relevant “characteristics”

Our response: Changed. The data is shown in newly submitted Table S1.

Table 2. Primers used in this study: Font and size

Our response: Changed.

Page 8: Section 2.3: Cells were cultured at 37 °C: Delete the bold for “C”

Our response: Changed.

Page 9: Section 2.7: Italicize “in vivo” from the first sentence.

Our response: Changed.

Page 10: Figure 2: Label (Font and size)

Our response: Changed. We carefully examined the labels for all Figs, set to Times New Roman, font 5, and bold. Please check the revised manuscript.

Page 11: Figure 3: Label (Font and size)

Our response: Changed. We carefully examined the labels for all Figs, set to Times New Roman, font 5, and bold. Please check the revised manuscript.

We suggest that the authors remove tables 1 and 2 and put them in the supplementary files.

Our response: Thanks for your advice. Tables 1 and 2 have been removed to the supplementary files, and named as Tables S1 and S2.

Reviewer 2 Report

Yan et al. investigated the role of VC1795 in intestinal colonization by V. cholerae. The authors showed that VC1795 is relevant for efficient mice colonization and is repressed by CRP. Furthermore, VC1795 negatively regulates the expression of  VC1794, which is negatively correlated with bacterial virulence. Although the study is interesting many parts of the manuscript are unintelligible and have to be revised. The section method needs a more detailed description.

1. The construction of mutant strains and complementation procedure needs more details.

2. Was the RNA treated with DNAse? Did authors run nRT control?

3. Experiments on mice require more details. How many mice were included in each group?

4. Why the expression gene data do not include the data for a complemented mutant?

5. Why no complementation was done for the VC1794 mutant?

The English style has to be improved.

Author Response

Reply Letter

Manuscript ID: ijms-2552480

Manuscript Title: The response regulator VC1795 of the VPI-2 contributes to intestinal colonization by Vibrio cholerae

Dear reviewer

Thank you very much for your valuable effort in handling this manuscript, ijms-2552480. We are also very grateful to you for your constructive comments, which helped us improve the quality of this research paper.

We hereby resubmit a revised manuscript to IJMS after making revisions and carefully considering all the comments. On the following pages, the detailed point-by-point responses to each comment are provided.

Thank you very much for your continued consideration of our work.

Best Regards,

Boyang Cao, Professor

Nankai University

Comments for Authors from reviewer 2 :

Yan et al. investigated the role of VC1795 in intestinal colonization by V. cholerae. The authors showed that VC1795 is relevant for efficient mice colonization and is repressed by CRP. Furthermore, VC1795 negatively regulates the expression of VC1794, which is negatively correlated with bacterial virulence. Although the study is interesting many parts of the manuscript are unintelligible and have to be revised. The section method needs a more detailed description.

Major Comments:

  1. The construction of mutant strains and complementation procedure needs more details.

Our response: Thanks for your advice. The relevant explanation was added in page 6, lines 113-121 and page 7, lines 122-129 of the revised manuscript.

  1. Was the RNA treated with DNAse? Did authors run nRT control?

Our response: Thank you for your attention. The RNA was treated with DNAse in this study. The relevant explanation was added in page 7, lines 135-136 of the revised manuscript. We did not run an nRT control; however, the 16S rRNA gene was used as a reference control for sample normalization.

  1. Experiments on mice require more details. How many mice were included in each group?

Our response: Thanks for your suggestion. The relevant explanation was added in page 10, lines 203-214 of the revised manuscript.

  1. Why the expression gene data do not include the data for a complemented mutant?

Our response: Thank you. Since the response regulator VC1795 was studied in the report, so most of the data, including the deletion mutation, qRT-PCR, pull-down assay, ChIP, and electrophoretic mobility shift assays (EMSAs), were collected and centered on both WT and Δvc1795. Well, it's a good suggestion to introduce the gene expression data of the complement strains in the future to make the data more comparable.

  1. Why no complementation was done for the VC1794mutant?

Our response: Thank you for your suggestion. In this study, the response regulator VC1795 was centered and studied, and its complementation of Δvc1795+pvc1795 was also constructed for the related experiments and data collection. VC1794 was found, however, 1) it's the downstream gene of VC1795; 2) it's not a regulator; 3) the effect of VC1794 for the other genes in the chromosome is uncertain. Hence, further efforts on VC1794 was less, but will be considered in the future to make the data more completely.

Round 2

Reviewer 1 Report

The authors have well adressed all my concerns

Author Response

Dear reviewer

Thank you very much for your valuable effort in handling our manuscript. We are also very grateful to you for your constructive comments, which helped us improve the quality of this research paper.

Best Regards,

Boyang Cao, Professor

Nankai University